# Occurrence Location and Propagation Inconformity Characteristics of Vibration Events in a Heading Face ofa Coal Mine

**DOI:** 10.3390/ijerph192215169

**Published:** 2022-11-17

**Authors:** Jianju Ren, Wenlong Zhang, Hongmei Zhang, Honggang Kou

**Affiliations:** 1School of Energy and Mining Engineering, China University of Mining and Technology (Beijing), Beijing 100083, China; 2School of Civil Engineering and Architecture, Qingdao Huanghai University, Qingdao 266427, China; 3Research Institute of Shaanxi Yanchang Petroleum (Group) Co., Ltd., Xi’an 710065, China; 4College of Civil Engineering, Xijing University, Xi’an 710123, China; 5Shanxi Key Laboratory of Safety and Durability of Concrete Structures, Xijing University, Xi’an 710123, China

**Keywords:** vibration events, rock burst, occurrence position, propagation characteristics, heading face, coal mine

## Abstract

The location and characteristics of the vibration event in the heading face of a coal mine are of great significance for the monitoring and early warning of rock burst. The aim of the study is to reveal the occurrence location and propagation characteristics of macro vibration events in a heading face of coal mine. After statistics and analysis, the occurrence location of the events is mostly around the head of heading face revealed by amplitude sequencing and arrival time sequencing. As the vibration event propagates to the rear sensors, the amplitude presents exponential attenuation, which is the same as the ideal state; however, the main frequency does not appear with linear attenuation, which is obviously different from the ideal state. The reason for the inconsistency of the main frequency is probably related to the complex underground environment. The results of the main frequency results in this study are completely opposite to the previous study, indicating that the inconsistency of the main frequency should be regarded carefully when using as an early warning index.

## 1. Introduction

Rock burst is a kind of dynamic disaster [1] with strong harmfulness in coal mines [2], which often causes hundreds of meters of roadway damage and dozens of casualties, and greatly affects the normal production of a coal mine [3]. At present, the main reason for the occurrence of rock burst accidents in China’s coal mines is that the mechanism [4] has not been clearly studied [5], leading to the poor monitoring, early warning [6], and governance measures [7,8]. Rock burst can be divided into a working face accident [9] and a heading face accident [10,11], according to the occurrence position; the academia is mostly focused on the working face accidents in the past period and ignores the heading face accidents; however, the studies [12] show that 40.74% of 108 counted accidents in the Yima ore district occurred in heading face, which promote researchers to pay more attention to the monitoring and early-warning method of rock burst in heading face [13]. The occurrences of a large number of high intensity vibration events are usually one of the precursors of rock burst accidents, to which attention should be paid. In addition, the adsorption relationship between gas and solid [14] may also have some influence.

The main difficulties of rock burst monitoring in heading face are as follows: (1) the mining impact caused by the heading procedure is small [15] and the energy source of such a serious accident injury [16] is not clear; (2) the relationship between the occurrence of the accident and the original rock stress (geological structure) [17] is not clear enough [18]; (3) the relationship between the occurrence of the vibration event and the advance abutment pressure [19] or fracturing [20] is not clear; (4) the occurrence location of the vibration event is not clear; (5) the research on the propagation law and spectrum characteristics of the vibration events [21] in heading face are rarely studied; (6) the sensor layout [22] can only be in a straight line in the heading face, which limits the precise location of vibration events; and (7) the heading speed substantially increases with the continuous progress of the heading equipment, resulting in the fact that the requirements for the sensor group shifting are higher. The above description shows that it is of great significance to solve the above problems for the successful monitoring and early warning in heading face. Among them, the location and spectrum propagation characteristics of vibration events are necessary to dig deeply through the vibration signal acquisition system, which can adapt to the heading face.

It is the most appropriate to use vibration events as index to reflect the risk of rock burst [23], and the vibration events in coal mines are mainly divided into roof breaking events [24] and coal fracture events [25]. In most cases, the heading face is not affected by the roof breaking events [26,27], due to the fact that it is usually far away from the mining face; thus, most of the macro vibration events are coal breaking events [25]. The on-site macro vibration events are different form acoustic emission events in laboratory tests [28,29]; the main difference between the two events lies in the size of coal fracture, and the resulting different energy, frequency, duration, sound [30], etc. In general, the larger the fracture range of coal, the greater the energy [23], the lower the dominant frequency [26,31], the longer the duration, the greater the sound [32], and the longer the propagation distance [33] of the vibration event. Another point worth noting is that the location closer to the event point will produce an earlier arrival time [34,35] and a greater amplitude [36,37]. Due to the mall energy of coal fracture, acoustic emission events are usually extremely weak [38], which need to be collected in the laboratory without interference [39]. However, there is much mechanical, electromagnetic, current, and other interference [40,41] in an actual coal mine, which results in the acoustic emission events being difficult to obtain correctly. Therefore, macro coal vibration events are usually used as monitoring indicators in the field [42,43].

Some laboratory results [44,45] and few field data results [46,47] show that the amplitude will decay exponentially and the dominant frequency will decrease linearly with the propagation of vibration events, which is theoretically correct. In recent years, some field application results [48,49] also use the change or deviation of the main frequency for monitoring and early warning. It should be noted that the law under the ideal state may be quite different from the actual conditions in heading face, especially for the target area with a complex construction environment. Whether the amplitude and main frequency of on-site vibration events in the heading face of a coal mine have ideal attenuation characteristics is directly related to the basis of monitoring. In addition, it is worth noting that most of the above studies are based on the statistics and analysis of a few events; individual cases cannot often represent all of them and relatively big data statistics are very necessary.

## 2. Methods

The data sources of the study are obtained though the arranged MS system in a gateway of Tengdong coal mine, Shandong Province, China. The monitoring system (as Figure 1) consists of four vibration sensors, an acquisition instrument, a ground server, and transmission cables. The sensor is used to receive the vibration signals and transmit them to the underground acquisition instrument through transmission cables. The acquisition instrument collects signals of the four sensors and transmits them to the ground server for collection and analysis. Four sensors are arranged in a straight line due to the limited space in heading face, although the arrangement causes the result that the events cannot be three-dimensionally located; it is more conducive to study the change of amplitude and frequency at different propagation distances to reflect one-dimensional positioning results. The daily heading speed of the heading face is 7~8 m; that is, the heading speed of a week is 49~56 m, and the four sensors move forward alternately every week on average to meet the signal acquisition with the heading face moving forward; thus, avoiding missing events due to an overly long propagation distance.

In order to collect vibration events more comprehensively, and simultaneously ensure the smooth process of transmission and storage, the sampling frequency of the acquisition instrument is designed as 2000 Hz. The sampling frequency can ensure the effective collection of field vibration events and meanwhile, match with the receiving frequency of the sensor (0~800 Hz). The acquisition instrument and sensor are made in China and can meet the safety requirements of China’s underground coal mines, including explosion-proof signs and safety signs, etc. The sensor is a kind of moving coil sensor, which does not need an additional power supply and saves a lot of trouble. The sensor is installed at the end of the roof bolt and the acquisition instrument is placed at the beginning of the heading face.

The designed distance between the four sensors is 50 m, and the distance between the most forward sensor and the heading face is 15 m, as shown in Figure 2. The four sensors are connected to different channels of the acquisition instrument, and the order of sensors is #1, #4, #3, #2 from front to back when they are initially installed. Here, in order to express the positional order of the sensors more conveniently, the simplified expression is named as 1432. The distance between #1 sensor and the heading face is increasing with the heading work, and when the distance reaches 65 m, the last #2 sensor moves to the most forward and the sensor sequence turns to 2143.

When the vibration signals occur, the four sensors will receive the vibration signal if the propagation distance is far enough, and the acquisition instrument will collect and store the signals. The arrival time and amplitude of the event can be used to determine the preliminary one-dimensional location of the vibration event (the smaller the propagation distance, the earlier the arrival time, the greater the amplitude). After the preliminary location, the frequency results of the four sensors are then analyzed according to different propagation distances. Through the above method, the preliminary one-dimensional position of the event, and the propagation law of amplitude and main frequency under different propagation distance conditions are well-studied under the field conditions.

## 3. Results

Table 1 shows the four different sensor arrangements periods (the periods are June 1~June 12, June 13~June 23, June 23~June 30, and July 1~July 9, respectively) of the analysis cycle; the corresponding sensor arrangements are, respectively, 1432, 2143, 3214, and 4321; that is, #2, #3, and #4 sensors are moved to the most forward of the heading face, respectively, on June 13, June 23, and July 1.

Ideally, sensors closer to the event location have a larger amplitude and an earlier arrival time. Thus, a case event waveform that occurred on July 4 (the arranged sensor sequencing is 4321) is presented in Figure 3; the results show that the magnitude sequencing and arrival time sequencing of the case event are all the same as the arranged sensor sequencing. The magnitude of #4 sensor is the biggest and the arrival time of #4 sensor is the earliest, which are consistent with the characteristics in an ideal state. The results show that the vibration signal acquisition system runs well; the method of amplitude sequencing and arrival time sequencing can be used to achieve the aim of the study.

The magnitude sequencing and arrival time sequencing of events that caused all four sensors to be affected in the four different periods are counted; the total number of statistical events is 338, and 89, 131, 84, and 34 events occurred in the four periods, respectively. Combined with the sensors arrangement sequencing, the preliminary magnitude sequencing and arrival time sequencing of statistical events are shown in Figure 4 and Figure 5, respectively. It can be preliminarily seen from the figures that the sequencing results of amplitude and arrival time are mostly the same as those of sensor arrangement in the four different stages; moreover, the most forward sensor always has the largest amplitude and the earliest arrival time, which indicates that most events occur in front of the most forward sensor and possibly around the head of heading face (the sequencing results in each period are not affected by the distance between the most forward sensor and the head of heading face). It should also be pointed out that the sequencing results of the amplitude and arrival time are basically the same (the correlation coefficient [50] is 0.911); and the two statistical results all confirm the result of event location. In contrast, the regularity of arrival time sequencing in the different stages is stronger than that of amplitude sequencing.

The identical proportion of amplitude sequencing, which is the same as the sensor arrangement sequencing in different stages, is counted in Figure 6. The statistical quantity of amplitude sequencing, which is the same as the sensor arrangement sequencing, is, respectively, 85, 105, 76, and 28 in the four periods; the corresponding proportion is, respectively, 96%, 80%, 90%, and 82%. In the same way, after statistics, the statistical quantity of arrival time sequencing, which is the same as the sensor arrangement sequencing, is, respectively, 85, 111, 72, and 32 in the four periods; the corresponding proportion is, respectively, 96%, 85%, 86%, and 94%. The statistical proportion results of amplitude sequencing and arrival time sequencing all show that most of (at least 82%) of the statistical vibration events occur in front of the most forward sensor. It should be stated that the reason for the small amount of vibration events in the fourth statistical stage (July 1 to July 9) is that the excavation work was not carried out for three days.

The above results about the location of vibration events provide an important basis for the amplitude and frequency changes of different propagation distances: the vibration events occur in front of the most forward sensor and propagate to the rear sensor through the same medium; and, theoretically, the amplitude and main frequency should all be attenuated. To verify this result, the statistical box line diagram of amplitude attenuation is shown in Figure 7 (here, in order to simplify the expression, the sensor sequence from near to far is all expressed as 4321); the results show that the amplitude presents an obvious attenuation trend with the increase of the propagation distance, and the amplitude attenuation rate of the front #4 sensor to #3 sensor is the largest, which is totally consistent with the theoretical result of amplitude attenuation. From the point distribution, the sensors closer to each other are more dispersed, which may be related to the propagation characteristics.

The statistical violin diagram of the main frequency attenuation of the same events is shown in Figure 8, and theoretically, the main frequency attenuation should show a linear decay; however, the statistical results show that there is no such rule at all—the average value of the main frequency does not decrease, but is increased. The average main frequency of the nearest #4 sensor is 81 Hz and the main frequency shows a high concentration. The average main frequency of the #3 sensor is reduced to 72 Hz and there are some new distributions around 400 Hz. The average main frequency of #2 sensor is 86 Hz, and the main frequency slightly increases compared with #3 and #4 sensors. The average main frequency of the last #1 sensor turns out to be 104 Hz and the main frequency substantially increases compared with the front three sensors; some higher main frequency distributions exit in #1 sensor, such as 270 Hz, 490 Hz, and 780 Hz. To sum up, the main frequency does not attenuate with the increase of propagation distance, and the farther sensor distribution is more dispersed in the frequency distribution.

## 4. Discussion

The preliminary location of the vibration event is obtained above; the amplitude and main frequency changes with the increasing propagation distance of the vibration event are studied in the “results” chapter. The results show that the vibration event occurs in front of the most forward sensor and the average amplitude of the statistical events show a decaying trend, which is consistent with the theoretical results; however, the main frequency of the statistical event does not show a decaying trend, which is not consistent with the theoretical results. For the amplitude indicator, the energy of vibration event is continuously absorbed by the same medium in the process of propagation, and the remaining energy is smaller, which leads to the decay of the amplitude. The main frequency represents the cumulative amplitude of the corresponding sine wave in a frequency that is the biggest; theoretically, the high-frequency component accounts for more when the distance is small; however, as the distance increases, the high-frequency component decays until it disappears, resulting in a relatively increased component of low frequency. The reason for the results of the field test are most likely that some low-frequency components are dispersed to a high frequency during the propagation process, due to the complexity of the underground environment (hidden joints, fissures, inhomogeneity of the medium, etc.), which results in a higher percentage of high-frequency components at greater distances.

In the “results” chapter, the statistical results about the amplitude decay are based on a large number of events, which respect the change in the mean and overall distribution of the statistical events; however, in fact, the results should also be counted and analyzed for each event to reflect the variation. Thus, the amplitude variation of each statistical event are counted in Figure 9. Events with different initial amplitudes all show a tendency to decay as the propagation distance increases; the difference lies in the fact that some amplitudes have a large decay and some have a small decay. However, in general, the number of events showing exponential decay accounts for the majority (only three statistical events present a small amplitude decay). Some events have similar laws, which may be related to their focal mechanism and propagation.

Following the same method, the variation of the main frequency with propagation distance for each event is counted in Figure 10. The initial main frequencies of the closest #4 sensor are basically concentrated in two interval ranges: 60~80 Hz and 330~340 Hz; the main frequencies of the following second-closest #3 sensor are basically concentrated in two interval ranges too. However, the bigger ranges change to 380~390 Hz, and it is worth noting that the higher frequency range of #3 sensor comes from the lower frequency range of the #4 sensor; the higher frequency range of #4 sensor all changes to a lower frequency range when propagating to #3 sensor; and the main frequencies of the following #2 sensor and #1 sensor become more decentralized, and concentration disappears. The main frequency variation of each statistical event indicates that the initial main frequency is concentrated; however, the main frequency becomes dispersed after propagation in the underground surrounding rock. In addition, the main frequency of most events did not show significant changes (basically varying from 50~100 Hz); other small numbers of events varied significantly and only three events showed significant decay. The variation results of the main frequency of each event demonstrate a large gap between the actual complex underground environment and the ideal state.

In order to present the change of the main frequency more graphically, two vibration event frequency waterfall plots with increasing and decreasing main frequencies are selected and shown in Figure 11 and Figure 12, respectively. It can be seen form Figure 11 that the main frequency of the nearest #4 sensor is 81 Hz; and the frequency goes through 71 Hz, 86 Hz, and finally becomes the 186 Hz of the farthest #1 sensor. In the process from near to far, both low-frequency components and high-frequency components are attenuated; however, the degree of attenuation is different, or some low-frequency components are transformed into high-frequency components, which result in the increase of the main frequency. In Figure 12, the main frequency changes are quite different; both low-frequency components and high-frequency components are all attenuated too. The difference of attenuation degree between the two components is smaller than that in Figure 12 and there is no transformation phenomenon of low-frequency components changing into high-frequency components.

This study obtained the preliminary one-dimensional location of macro vibration events in a heading face of a coal mine, and statistically analyzed the results of amplitude and main frequency changes with propagation distance based on the occurrence location of vibration events. The results showed that the vibration events occurred in front of the most forward sensor and the on-site amplitude changes are fully consistent with the changes of the ideal state; however, the changes of main frequency reflected a completely different situation from the ideal state; the reason for this phenomenon has a great relationship with the complex environment of the underground coal mine. The innovative point of the study is to reveal the gap between main frequency results of the actual field and the ideal state, and this result is far from the other research results. There may not be such a rule in the heading face in other places; in the future research and application process, researchers should carefully use the main frequency index, because its performance in the field is completely different. The research results of this paper are obtained only from a heading face case; more on-site verification work is needed in the future study.

## 5. Conclusions

Underground vibration events are often used as the main object of rock burst monitoring and early warning; in addition, the study about its occurrence location and propagation characteristics in heading face is particularly meaningful. Based on a large number of field data, the preliminary one-dimensional occurrence location and amplitude frequency propagation characteristics of vibration events in heading face are counted and analyzed. The main innovation of this paper is that it successfully reveals the preliminary occurrence location of the vibration event; and what is more, it obtains the conclusion that the main frequency of the on-site vibration event does not decay at all, which is completely opposite to the ideal state and other previous research results. In addition, the results provide an important basis for the early warning index of rock burst monitoring in heading face.

## Figures and Tables

**Figure 1 ijerph-19-15169-f001:**
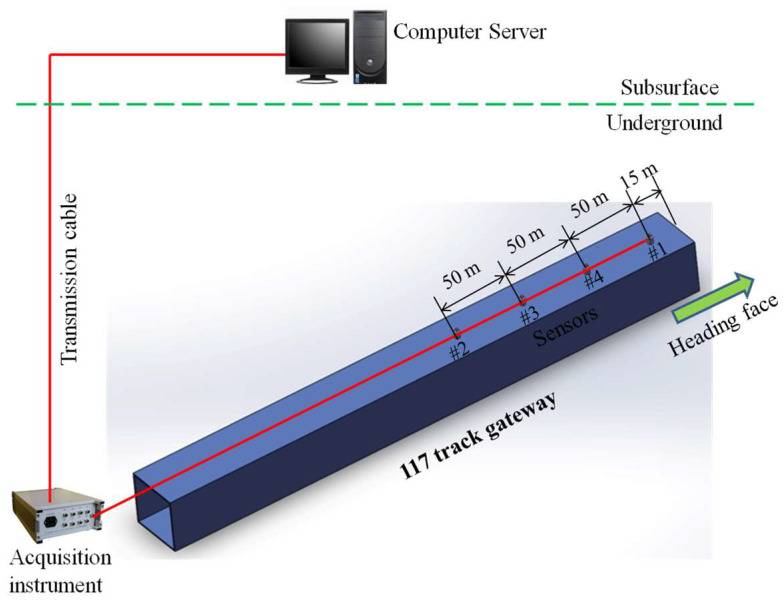
Monitoring system installed in the gateway of Tengdong coal mine.

**Figure 2 ijerph-19-15169-f002:**
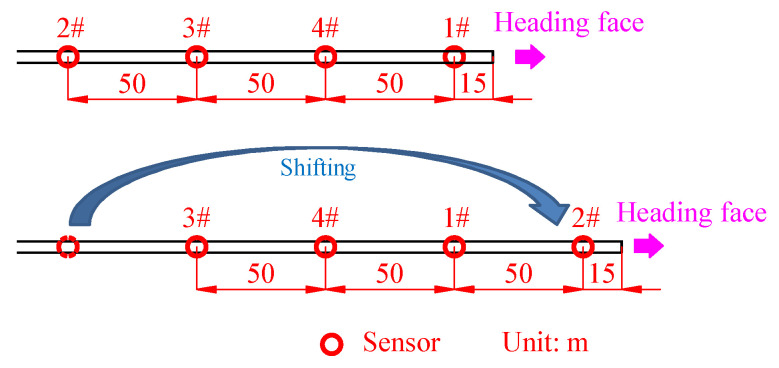
Sensor movement rules.

**Figure 3 ijerph-19-15169-f003:**
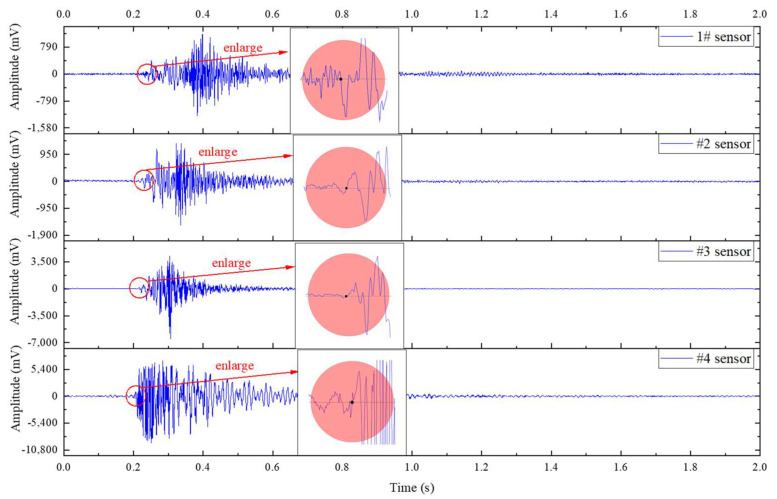
A case event waveform to show magnitude sequencing and arrival time sequencing.

**Figure 4 ijerph-19-15169-f004:**
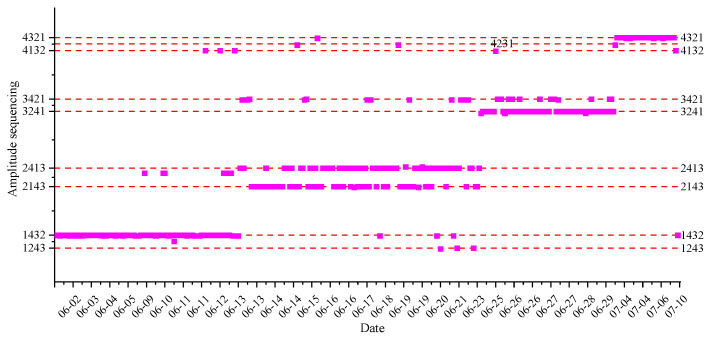
Amplitude sequencing results in the four different periods.

**Figure 5 ijerph-19-15169-f005:**
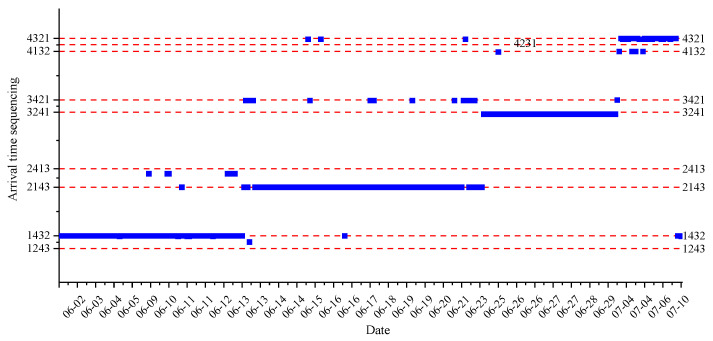
Arrival time results in the four different periods.

**Figure 6 ijerph-19-15169-f006:**
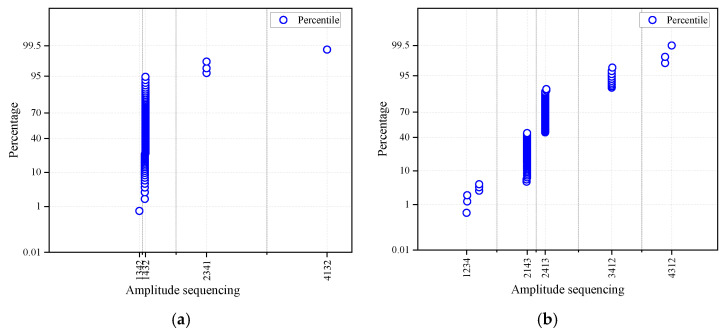
Identical statistical proportion of amplitude sequencing in four periods. (**a**) June 1~June 12; (**b**) June 13~June 23; (**c**) June 23~June 30; (**d**) July 1~July 9.

**Figure 7 ijerph-19-15169-f007:**
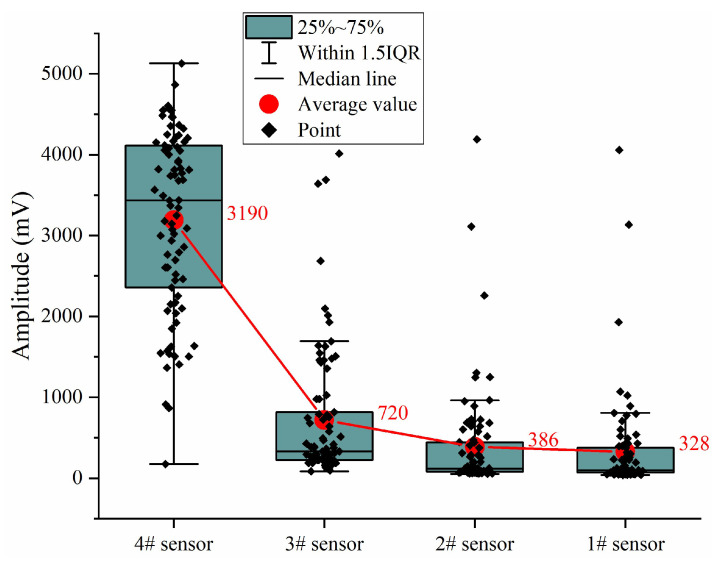
Statistical box line diagram of amplitude attenuation.

**Figure 8 ijerph-19-15169-f008:**
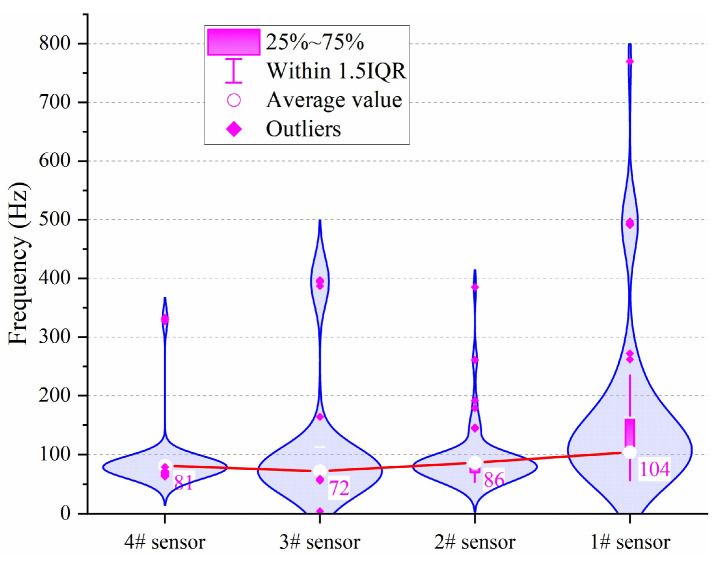
Statistical violin diagram of main frequency.

**Figure 9 ijerph-19-15169-f009:**
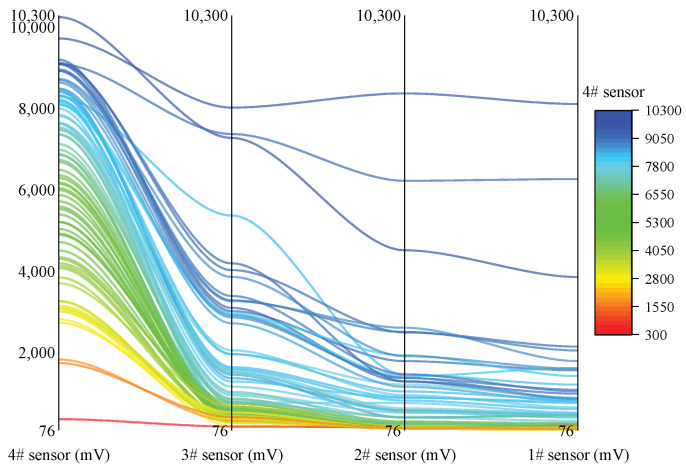
Amplitude variation of each statistical event.

**Figure 10 ijerph-19-15169-f010:**
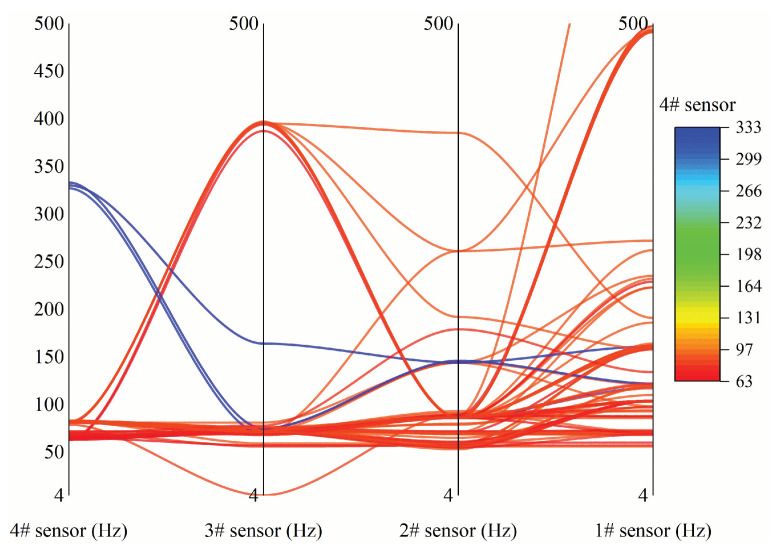
Main frequency variation of each statistical event.

**Figure 11 ijerph-19-15169-f011:**
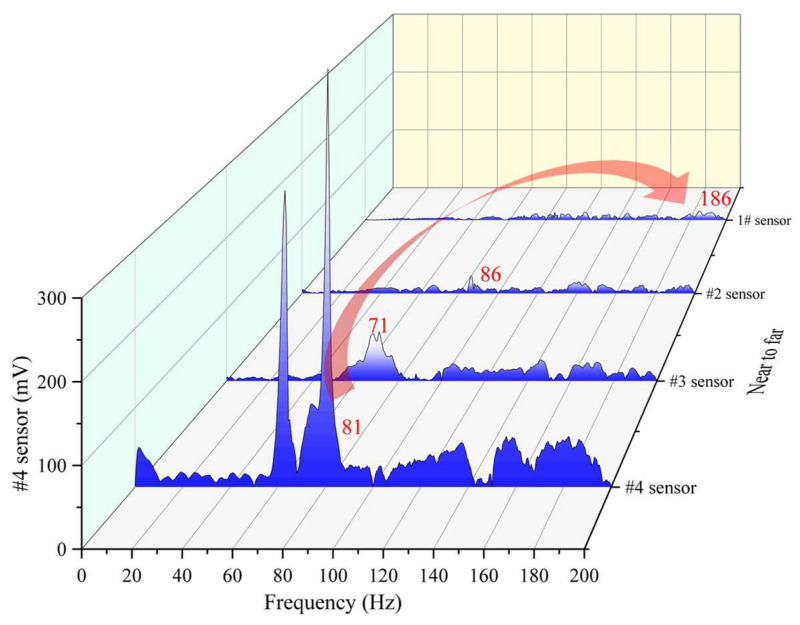
Main frequency increasing case event.

**Figure 12 ijerph-19-15169-f012:**
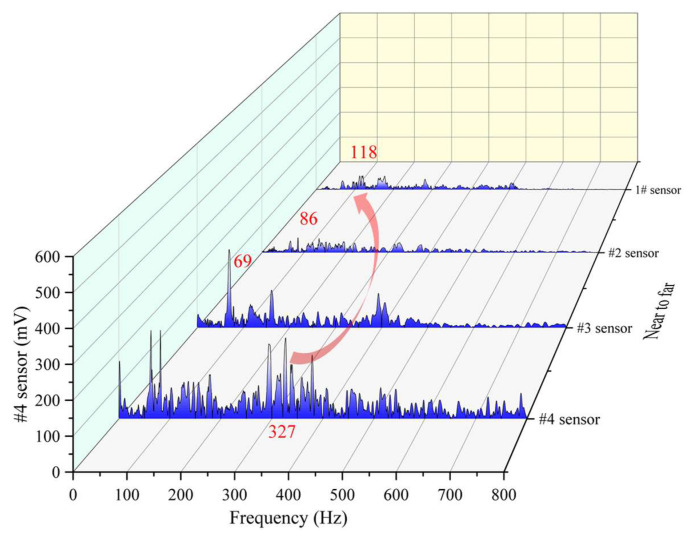
Main frequency decreasing case event.

**Table 1 ijerph-19-15169-t001:** Sensor arrangements in four different periods.

Periods	Sensor Arrangement	Shifting Sensor
June 1~June 12	1432	
June 13~June 23	2143	#2
June 23~June 30	3214	#3
July 1~July 9	4321	#4

## Data Availability

Not applicable.

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
