# Peer review of "Occurrence Location and Propagation Inconformity Characteristics of Vibration Events in a Heading Face ofa Coal Mine"

_ijerph, 2022, doi:10.3390/ijerph192215169_

Round 1

Reviewer 1 Report

For the paper “Occurrence Location and Propagation Inconformity Characteristics of Vibration Events in a Heading Face of Coal Mine”, it has a good structure and data analysis method, and has obtained good results. The main frequency does not appear linear attenuation, which is obviously different from the ideal state. Before publication, the following minor issues should be solved:

1. Optimize the Abstract to make its structure more reasonable and highlight the focus of this study;

2. In the first paragraph of the introduction chapter, the authors wrote about the basic situation of rock burst, but did not explain the relationship between the vibration events and rock burst in this paper;

3. For “Figure 4. Amplitude sequencing results in the four different periods” and “Figure 5. Arrival time results in the four different periods”, some boundaries of different stages should be added to better reflect the sensor sequencing of different stages;

4. For the description of “Figure 9. Amplitude variation of each statistical event”, the authors should add the comparison of change trends of different events;

5. The research results of this paper are obtained only from a heading face case. The authors should make it clear that there may not be such a rule in the heading face in other places to reflect the actual situation and some limitations of this study;

6. Seek a native English speaker to help check the grammar of the article.

Author Response

  1. Thank you for your excellent comments, we add the contents in the Abstract section: “The location and characteristics of the vibration event in the heading face of coal mine are of great significance for the monitoring and early warning of rock burst”;
  2. Thank you for your excellent comments, we add the contents in the first paragraph of the introduction chapter: “The occurrences of a large number of high intensity vibration events are usually one of the precursors of rock burst accidents, which should be paid attention to”;
  3. Thank you for your excellent comments, but the dots and lines in our figure show obvious stage characteristics, and we don't think it is necessary to add some symbolic signs. We hope to get your understanding.
  4. Thank you for your excellent comments, we add the contents in the description of “Figure 9. Amplitude variation of each statistical event”: “Some events have similar laws, which may be related to their focal mechanism and propagation”;
  5. we add the contents in the last paragraph of the discussion section: “There may not be such a rule in the heading face in other places, in the future research and application process, researchers should carefully use the main frequency index, because its performance in the field is completely different”;
  6. Thank you for your reminding. We have found a professional to help us read and revise the article. Thank you.

Reviewer 2 Report

The aim of the study is to reveal the occurrence location and propagation characteristics of macro vibration events in a heading face of coal mine. Results show that the occurrence location of the events is mostly around the head of heading face. Interestingly, the amplitude presents exponential attenuation, which is the same as the ideal state, but the main frequency does not appear linear attenuation, which is obviously different from the ideal state. The results of the main frequency results in this study are opposite to the previous study, which reflect the particularity of the generation environment and propagation of underground events in the heading face. The overall structure and conclusions of the article are appropriate, but I think the following modifications should be made before acceptance:

(1) In the Abstract section, the authors stated that “the aim of the study is to reveal the occurrence location and propagation characteristics of macro vibration events in a heading face of coal mine”. I suggest that the importance of the research on this aspect should be added before the description, and then the goal of this paper should be put forward, which will be more consistent with the writing rules.

(2) In the Introduction section, the authors stated that “the above description shows that it is of great significance for the successful monitoring and early warning to obtain the location and spectrum propagation characteristics of vibration events through the vibration signal acquisition system which can adapt to the heading face”, the description can be simplified and added to the Abstract section. In addition, the main difficulties of rock burst monitoring in heading face are expounded, but after the difficulties are stated, a summary discourse should be written to cohere the theme of this paper, like: “the above description shows that it is of great significance to solve the above problems for the successful monitoring and early warning in heading face. Among them, the location and spectrum propagation characteristics of vibration events are necessary to dig deeply through the vibration signal acquisition system which can adapt to the heading face”.

(3) I suggest that the expression of “When the vibration signals occur, the four sensors will receive the vibration signal if the propagation distance is far enough and the signals are then transmitted to the acquisition instrument” in the last paragraph of the Methods section can be replaced as “When the vibration signals occur, the four sensors will receive the vibration signal if the propagation distance is far enough and the acquisition instrument will collect and store the signals”.

(4) The Figure 3 should be beautified. Some words in the picture feel flat.

(5) Figure 7 focuses on the change of mean value, but does not analyze the distribution of points, and this aspect should be added.

(6) “Figure 11. Main frequency increasing event” can be changed to “Figure 11. Main frequency increasing case event”. And the same as Figure 12.

(7) Read the full text to avoid some small spelling mistakes and grammar problems.

Author Response

(1) Thank you for your excellent comments, we add the contents in the Abstract section: “The location and characteristics of the vibration event in the heading face of coal mine are of great significance for the monitoring and early warning of rock burst”;

(2) Thank you for your reminding. We have revised it as you said. See the revised manuscript for details;

(3) Thank you for your reminding. We have replaced it as you said. See the revised manuscript for details;

(4) We have adjusted it as you said. See the revised manuscript for details;

(5) We added the contents in the the distribution of Fig. 7: “From the point distribution, the sensors closer to each other are more dispersed, which may be related to the propagation characteristics”;

(6) Thank you for your reminding. We have revised it as you said. See the revised manuscript for details;

(7) Thank you for your reminding. We have found a professional to help us read and revise the article. Thank you.